# Reconciling Kaplan and Chinchilla Scaling Laws

**Tim Pearce** *Microsoft Research*

**Jinyeop Song** *MIT*

**Reviewed on OpenReview:** *https://openreview.net/forum?id=NLoaLyuUUF*

## Abstract

Kaplan et al. (2020) ('Kaplan') and Hoffmann et al. (2022) ('Chinchilla') studied the scaling behavior of transformers trained on next-token language prediction. These studies produced different estimates for how the number of parameters ($N$) and training tokens ($D$) should be set to achieve the lowest possible loss for a given compute budget ($C$). Kaplan: $N_{\text{optimal}} \propto C^{0.73}$, Chinchilla: $N_{\text{optimal}} \propto C^{0.50}$. This paper finds that much of this discrepancy can be attributed to Kaplan counting non-embedding rather than total parameters, combined with their analysis being performed at small scale. Simulating the Chinchilla study under these conditions produces biased scaling coefficients close to Kaplan's. Hence, this paper reaffirms Chinchilla's scaling coefficients, by explaining the primary cause of Kaplan's original overestimation. As a second contribution, the paper explains differences in the reported relationships between loss and compute. These findings lead us to recommend that future scaling studies use total parameters and compute. [1]

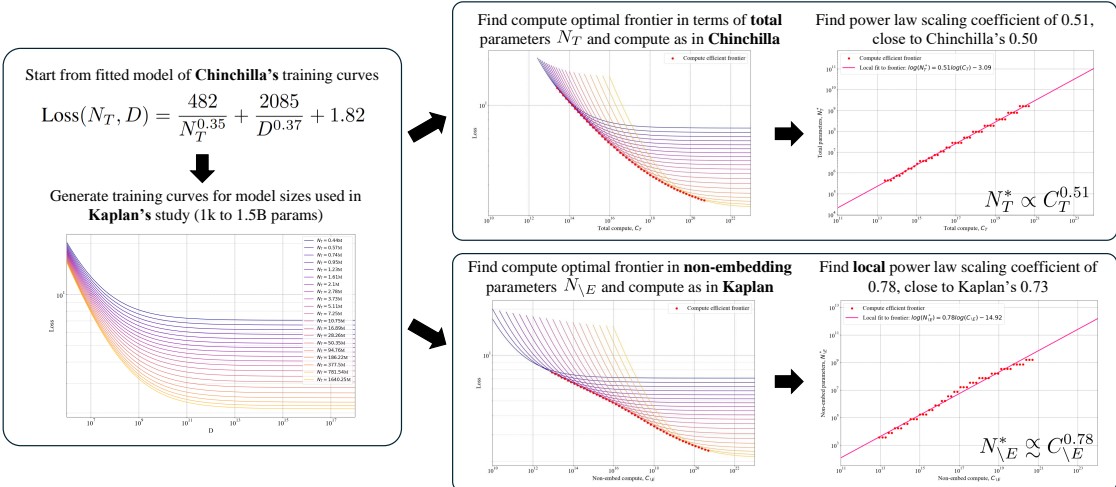

Figure 1: Overview of the approach used to reconcile the two studies.

## 1 Introduction

Kaplan et al. (2020) ('Kaplan') and Hoffmann et al. (2022) ('Chinchilla') provided two influential studies measuring the impact of scale in large language models (LLMs). Both informed large-scale efforts on how to trade off model parameters ($N$) and training tokens ($D$) for a given compute budget ($C$), but with conflicting advice. Kaplan's finding that $N_{\text{optimal}} \propto C^{0.73}, D_{\text{optimal}} \propto C^{0.27}$ led to the conclusion that *"big models may be more important than big data"*, and LLMs trained in the ensuing years committed more resources to

---

[1]Code for analysis: https://github.com/TeaPearce/Reconciling_Kaplan_Chinchilla_Scaling_Laws

model size and less to data size. The subsequent Chinchilla study found $N_{\text{optimal}} \propto C^{0.50}, D_{\text{optimal}} \propto C^{0.50}$, leading to their main thesis *"for many current LLMs, smaller models should have been trained on more tokens to achieve the most performant model"*, sparking a trend towards LLMs of more modest model sizes being trained on more data.

**What was the cause of the difference in these scaling coefficient estimates that led to vast amounts of compute (plus emissions and finances) being used inefficiently?** There have been suggestions that differences could be explained by different optimization schemes (Hoffmann et al., 2022) or datasets (Bi et al., 2024). This note finds these suggestions incomplete, and offers a simple alternative explanation; much of the discrepancy can be attributed to Kaplan counting non-embedding rather than total parameters, combined with their analysis being performed at small scale.

We find that this methodological difference also plays a part in differences in the reported relationship between compute and loss. Kaplan: $L_{\text{optimal}} \propto C^{0.057}$, Chinchilla: $L_{\text{optimal}} - E \propto C^{0.178}$.

Concretely, this paper offers the following contributions.

- We develop an analytical approach comparing the scaling relationships reported in Kaplan and Chinchilla (Section 3). This approach finds that Kaplan's reported relationship is *locally* consistent with Chinchilla's, if non-embedding parameters are used, and at smaller scale.

- Secondly, we study the reported relationships between compute and loss (Section 5). Again non-embedding parameters and smaller scale models are the cause of Kaplan's biased estimate, combined with the absence of an offset term in their compute-loss form.

- We recommend that going forward the scaling community measures **total parameters** and **total compute**, and uses an offset in the compute-loss relationship.

## 2 Preliminaries

This section introduces some background knowledge and definitions (Section 2.1), outlines the analysis approach for our main result (Section 2.2), and documents assumptions (Section 2.3).

### 2.1 Set Up

**Neural scaling laws.** Kaplan et al. (2020) & Hoffmann et al. (2022) investigated empirically modeling the relationships between the number of parameters ($N$), training tokens ($D$), training compute ($C$) and loss ($L$) in transformers for language modeling. The main functional form of relationship considered was a power law, $y = ax^b$, which is widely used throughout the sciences to describe a relationship between two quantities ($x$ & $y$) that holds over many orders of magnitude, e.g. (Kello et al., 2010).

**Definition of $N$ & $C$.** One difference between the two studies is the definition of $N$ & $C$. Kaplan studied relationships in terms of non-embedding parameters ($N_{\backslash E}$) and non-embedding compute ($C_{\backslash E}$), excluding the contributions of the embedding layers for the vocabulary and position indices ($N_E$). By contrast, Chinchilla studied total parameters ($N_T$) and total compute ($C_T$). We define,

$$N_T = N_E + N_{\backslash E}, \tag{1}$$
$$N_E = (h + v)d, \tag{2}$$

where $d$ is the dimension of the transformer residual stream, $v$ is vocab size, $h$ is context length (only included when positional embeddings are learned). Using the common approximation for training compute FLOPs $C = 6ND$ (a factor of 6 covers a forward and backward pass), we define total and non-embedding compute,

$$C_T = 6N_T D = 6(N_E + N_{\backslash E})D, \tag{3}$$
$$C_{\backslash E} = 6N_{\backslash E}D. \tag{4}$$

**Compute optimality.** This definition of compute $C = 6ND$ suggests a direct trade off between parameters and training tokens for a given compute budget. The two studies focus on 'compute optimal' configurations of these quantities. That is, for a given compute budget, the parameters and training tokens that lead to the lowest possible loss. For total parameters this is written (using $*$ to signify 'optimal'),

$$N_T^* = \operatorname*{argmin}_{N_T \text{ s.t. } C_T = 6N_T D} L(N_T, C_T). \tag{5}$$

Given this notation, the estimated scaling coefficients can be more precisely written,

$$\text{Kaplan: } N_{\backslash E}^* \propto C_{\backslash E}^{0.73}, \tag{6}$$

$$\text{Chinchilla: } N_T^* \propto C_T^{0.50}. \tag{7}$$

(Note that whilst this work focuses on the scaling coefficient for parameters, by subscribing to $C = 6ND$ the data coefficient is implied; $N \propto C^a \implies C/D \propto C^a \implies D \propto C^{1-a}$.)

**Functional form.** An important functional form relating $N_T$, $D$, $L$ used in Chinchilla is given by,

$$L(N_T, D) = \frac{N_c}{N_T^\alpha} + \frac{D_c}{D^\beta} + E, \tag{8}$$

where $N_c, D_c, \alpha, \beta > 0$ are empirically fitted constants and $E$ is the 'irreducible' loss inherent in language. This form conveniently gives rise to power law relationships; $N_T^* \propto C_T^a$ with $a = \frac{\beta}{\alpha+\beta}$, $D_T^* \propto C_T^b$ with $b = \frac{\alpha}{\alpha+\beta}$, and $L_T^* - E \propto C_T^{-\gamma}$ with $\gamma = \frac{\alpha\beta}{\alpha+\beta}$.

There are two possible specifications from the constants in Eq. 8 – those originally reported in Chinchilla, and those reported in a re-analysis conducted by Besiroglu et al. (2024) claiming to correct minor errors in the fitting procedure. Our work reports results with both specifications.

$$\text{Chinchilla specification: } N_c = 406.4, Dc = 410.7, \alpha = 0.3392, \beta = 0.2849, E = 1.693 \implies N_T^* \propto C_T^{0.46}, \tag{9}$$

$$\text{Epoch AI specification: } N_c = 482.0, Dc = 2085.43, \alpha = 0.3478, \beta = 0.3658, E = 1.817 \implies N_T^* \propto C_T^{0.51}. \tag{10}$$

## 2.2 Analysis Overview

Our analysis uses information and data from the Chinchilla and Kaplan studies to estimate the scaling laws that would emerge if the Chinchilla relationship had been expressed in terms of $N_{\backslash E}$ & $C_{\backslash E}$, and this had been done over the smaller model sizes used in Kaplan, as summarized in Figure 1.

We will see that for large $N_T$, $N_E$ becomes a negligible portion of the model's parameters and compute cost. Hence in the large parameter regime the two coefficients directly conflict with each other. At smaller values of $N_T$, $N_E$ is *not* negligible (this is the regime considered in Kaplan's study – 768 to 1.5B parameters). We find that at the smaller end of this range, the relationship between $N_{\backslash E}^*$ & $C_{\backslash E}$ is not in fact a power law. However, fitting a "local" power law at this small scale, produces a coefficient that is close to Kaplan's, and hence roughly reconciles these two results.

Our approach in Section 3 is broken down as follows.

- **Step 1.** Fit a suitable function predicting $N_{\backslash E}$ from $N_T$.

- **Step 2.** Incorporate this function into a model predicting loss in terms of $N_T$ & $C_T$.

- **Step 3.** Analytically derive the relationship between $N_{\backslash E}^*$ & $C_{\backslash E}$.

- **Step 4.** Simulate synthetic data from the Chinchilla loss model over the model sizes used Kaplan. Fit a local power law for $N_{\backslash E}^*$ in terms of $C_{\backslash E}$.

Section 4 experimentally verifies our analysis by training a set of language models at tiny scale and conducting scaling law analyses under various settings. Simply changing the basis $N_T$ to $N_{\backslash E}$ produces coefficients inline with Chinchilla and Kaplan respectively, while multiple token budgets and decay schedules does not.

Section 5 presents a second, related contribution. We reconcile differences in the relationship between loss and compute proposed by the two studies. We leverage a similar analysis as above, but Step 3 & 4 are now performed with respect to the relationship between the optimal loss $L_{\backslash E}^*$, and compute $C_{\backslash E}$. So again, we begin from Chinchilla data, and correct for the exclusion of embedding parameters and compute, combined with the smaller model sizes used in Kaplan's study, and additionally a differing choice of fitting function. Through these corrections, we are able to roughly recover Kaplan's compute-loss coefficient, and hence reconcile the two studies.

### 2.3  Assumptions

For transparency, we list the assumptions and approximations made in our analysis.

- We assume $C_{\backslash E} = 6N_{\backslash E}D$ and $C_T = 6N_TD$.

- We assume a fixed functional form between total and non-embedding parameters in Eq. 11, and fit $\omega$ empirically using Chinchilla model configurations.

- We assume a fixed functional form between loss, total parameters and training data given by Eq. 8. We report results using both the Chinchilla (Eq. 9) and Epoch AI (Eq. 10) fitted constants.

- We approximate Kaplan's models with 20 logarithmically spaced model sizes from 0.79k to 1.58B non-embedding parameters.

## 3  Analysis: Compute-Parameter Scaling Coefficient

This Section presents our main analysis. We show that a *local* scaling coefficient of 0.74 to 0.78 (close to Kaplan's 0.73) can arise when computed in terms of non-embedding parameters in the small-parameter regime, whilst being consistent with Chinchilla's coefficient.

**Step 1.** *Fit a suitable function predicting $N_{\backslash E}$ from $N_T$.*

We require a suitable function relating non-embedding and total parameters. We propose to use the form

$$N_T = N_{\backslash E} + \omega N_{\backslash E}^{1/3}, \tag{11}$$

for some constant $\omega > 0$. Aside from having several nice properties (strictly increasing and $\lim_{N_T \to \infty} N_T = N_{\backslash E}$ [2]), it can be motivated from both the Kaplan and Chinchilla study.

**Kaplan perspective.** Consider Kaplan's method for parameter counting,

$$N_T = 12ld^2 + N_E, \tag{12}$$

where $l$ is number of layers. Whilst Kaplan do not list their model configurations, they do study varying aspect ratio $A = d/l$ for a fixed size model. They find that models of a given size perform similarly over a range of aspect ratios, and this is not affected by model scale (their Figure 5). Hence, we could assume a sizing scheme with fixed aspect ratio ($A \approx 40$ appears sensible from their plots). Assuming this sizing allows us to state (with $l = d/A$ in Eq. 12),

$$N_T = \frac{12}{A}d^3 + N_E. \tag{13}$$

---

[2]**Proof.** $N_T = N_{\backslash E} + \omega N_{\backslash E}^{1/3} \implies N_T/N_{\backslash E} = 1 + \omega N_{\backslash E}^{-2/3}$. Examining the r.h.s., $\lim_{N_T \to \infty} 1 + \omega N_{\backslash E}^{-2/3} = 1$, hence we conclude on the l.h.s $\lim_{N_T \to \infty} N_{\backslash E}/N_T = 1$ or $N_{\backslash E} = N_T$.

Observing that $N_{\backslash E} = (12/A)d^3 \implies d = (N_{\backslash E}(A/12))^{1/3}$, and combining with $N_E = (v+h)d$,

$$N_T = N_{\backslash E} + (v+h)\left(\frac{A}{12}\right)^{1/3} N_{\backslash E}^{1/3}. \tag{14}$$

This is the same form as Eq. 11 with $\omega = (v+h)\left(\frac{A}{12}\right)^{1/3}$.

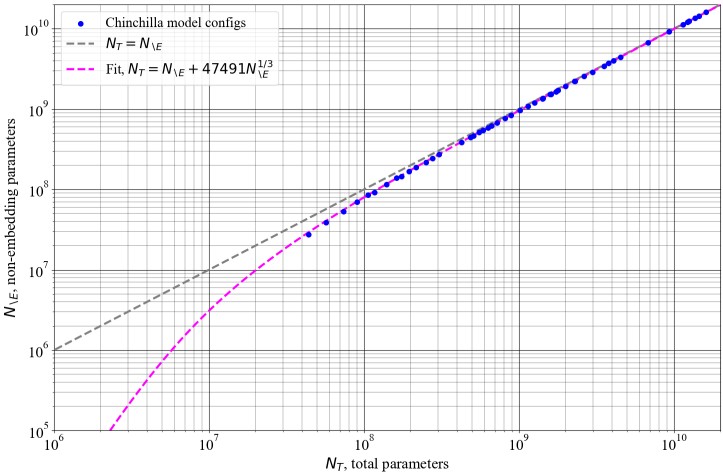

Figure 2: Total parameter count vs. non-embedding parameter count for the suite of models sizes used in the Chinchilla study, along with our fitted approximation. Note the curvature at model sizes below around 200M parameters.

**Chinchilla perspective.** We empirically fit a function $N_T = N_{\backslash E} + \omega N_{\backslash E}^{\delta}$ (note the learnable exponent) to the Chinchilla model configurations listed in Table A9 of Hoffmann et al. (2022) for a range of $N_T$ (44M to 16B). We compute $N_E$ from Eq. 2, using the reported vocab size of 32,000, but ignore the context length 2,048 since Chinchilla used non-learnable position embeddings (though their inclusion effects coefficients only slightly).

Figure 2 shows the configurations and relationship from a model fitted with numpy's `polyfit`, which produces coefficients, $\omega = 47491$ & $\delta = 0.34$. The exponent has come out close to $1/3$, and an implied aspect ratio $A = 39.2$ (inferred from $\omega$). Hence, this further supports the form in Eq. 11.

**Step 2.** *Incorporate this function into a model predicting loss in terms of $N_T$ & $C_T$.*

Recall that whilst we are interested in the dependence of $N_T^*$ on $C_T$, this arises only via their mutual relationship with loss

$$N_T^* = \underset{N_T \text{ s.t. } C_T = 6N_T D}{\operatorname{argmin}} L(N_T, C_T). \tag{15}$$

In order to analytically study their scaling relationship, we need an analytical form of loss, for which we use the functional form in the Chinchilla study. Taking Eq. 8 and using $C_T = 6N_T D$,

$$L(N_T, C_T) = \frac{N_c}{N_T^\alpha} + \frac{D_c}{(C_T/6N_T)^\beta} + E. \tag{16}$$

By differentiating Eq. 16 wrt $N_T$ and setting to zero, then rearranging in terms of $N_T$ we find

$$N_T^* = \left(\frac{\alpha}{\beta}\frac{N_c}{D_c}\right)^{\frac{1}{\alpha+\beta}} \left(\frac{C_T}{6}\right)^{\frac{\beta}{\alpha+\beta}} \text{ or simply } N_T^* \propto C^{\frac{\beta}{\alpha+\beta}}. \tag{17}$$

We now modify Eq. 16 to be in terms of non-embedding parameters and compute. Note whilst $N_T$ requires Eq. 11 from step 1, the second term avoids this as $D = C_T/6N_T = C_{\backslash E}/6N_{\backslash E}$.

$$L(N_{\backslash E}, C_{\backslash E}) = \frac{N_c}{(N_{\backslash E} + \omega N_{\backslash E}^{1/3})^\alpha} + \frac{D_c}{(C_{\backslash E}/6N_{\backslash E})^\beta} + E \tag{18}$$

**Step 3.** *Analytically derive the relationship between $N_{\backslash E}^*$ & $C_{\backslash E}$.*

To find the relationship between $N_{\backslash E}^*$ & $C_{\backslash E}$ we take the derivative of Eq. 18 wrt $N_{\backslash E}$, set to zero and rearrange,

$$6N_{\backslash E}^* \left(N_{\backslash E}^* + \frac{\omega}{3}N_{\backslash E}^{*1/3}\right)^{-\frac{1}{\beta}} \left(N_{\backslash E}^* + \omega N_{\backslash E}^{*1/3}\right)^{\frac{1+\alpha}{\beta}} \left(\frac{\beta}{\alpha}\frac{D_c}{N_c}\right)^{\frac{1}{\beta}} = C_{\backslash E}. \tag{19}$$

This shows that in general the relationship between $N_{\backslash E}^*$ & $C_{\backslash E}$ is *not* a power law. However, we can consider a "local" power law approximation. That is, for some particular value of $N_{\backslash E}$, there is some constant $g$ giving a first order approximation (denoted by $\stackrel{\propto}{\sim}$) $N_{\backslash E}^* \stackrel{\propto}{\sim} C_{\backslash E}^g$, where $g$ is defined

$$\frac{1}{g} := \frac{d\log(C_{\backslash E})}{d\log(N_{\backslash E}^*)} = 1 - \frac{1}{\beta}\frac{N_{\backslash E}^{*2/3} + \frac{\omega}{9}}{N_{\backslash E}^{*2/3} + \frac{\omega}{3}} + \frac{\alpha+1}{\beta}\frac{N_{\backslash E}^{*2/3} + \frac{\omega}{3}}{N_{\backslash E}^{*2/3} + \omega}. \tag{20}$$

Working is given in Appendix A.1. Figure 3 plots Eq. 19 & 20, using coefficients for $\alpha, \beta, N_c, D_c$ from the Epoch AI specification, and $\omega = 47491$. There are three phases.

- At small scale, $\lim_{N_{\backslash E}^* \to 0} \frac{1}{g} = \frac{\alpha/3+\beta}{\beta} \implies N_{\backslash E}^* \stackrel{\propto}{\sim} C_{\backslash E}^{\frac{\beta}{\alpha/3+\beta}}$ [3].

- At large scale, $\lim_{N_{\backslash E}^* \to \infty} \frac{1}{g} = \frac{\alpha+\beta}{\beta} \implies N_{\backslash E}^* \stackrel{\propto}{\sim} C_{\backslash E}^{\frac{\beta}{\alpha+\beta}}$ [4], as in the $N_T$ case in Eq. 17.

- There is also a transition phase, where $g$ briefly increases. This happens in between the two limits, when $N_{\backslash E}^{2/3}$ is of the same order as $\omega$. Indeed at exactly the point $N_{\backslash E}^{2/3} = \omega$, we have $N_T = N_{\backslash E} + \omega N_{\backslash E}^{1/3} = N_T = 2N_{\backslash E}$, or a 50/50 split between embedding and non-embedding parameters. In Figure 3 we see this transition region occurs around this point; $N_{\backslash E} = \omega^{3/2} = 47491^{3/2} \approx 1 \times 10^7$.

**Step 4.** *Simulate synthetic data from the Chinchilla loss model over the model sizes used Kaplan. Fit a local power law for $N_{\backslash E}^*$ in terms of $C_{\backslash E}$.*

By reading $g$ off Figure 3, we could estimate a local power law and hence scaling coefficient for a given value of $N_{\backslash E}^*$. However, it's not clear what $N_{\backslash E}^*$ point value is representative of the Kaplan study. We opt for a more faithful estimation procedure, generating synthetic training curves from Eq. 18 across the *range* of model sizes used in Kaplan, and fit coefficients using models falling on the compute efficient frontier. This will also verify our analytic expression for $N_{\backslash E}^*$ & $C_{\backslash E}$ in Eq. 19.

Figure 4 shows the synthetic training curves generated. We simulated 20 models with $N_{\backslash E}$ ranging from 790 parameters to 1.58B (Kaplan reports using model sizes *"ranging in size from 768 to 1.5 billion non-embedding parameters"*). For other constants in Eq. 18, we adopt the Epoch AI specification (Eq. 10) and $\omega = 47491$, though we report final results for the Chinchilla specification (Eq. 9) also.

**Main result.** Figure 5 shows the estimated scaling coefficient when fitting a power law to the compute optimal frontier (Chinchilla's Method 1) produced by these synthetic training curves. This marks our main

---

[3]**Proof.** As $N_{\backslash E}^* \to 0$, we can ignore $N_{\backslash E}^*$ terms and $1/g = 1 - (1/\beta)(3/9) + (\alpha+1)/3\beta = 1 + \alpha/3\beta$.

[4]**Proof.** As $N_{\backslash E}^* \to \infty$, we can ignore $\omega$ terms and $1/g = 1 - (1/\beta) + (\alpha+1)/\beta = 1 + \alpha/\beta$.

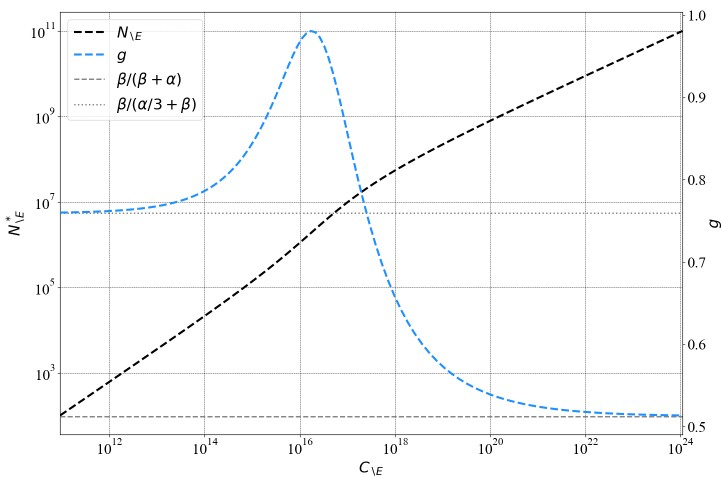

Figure 3: Visualization of Eq. 19 & 20, using the Epoch AI specification.

result – beginning from a model taken from Chinchilla's study, and modifying two aspects to align with Kaplan's study ($N_T \to N_{\setminus E}$, small model sizes 0.79k – 1.58B parameters), we find *local* scaling coefficients,

$$\text{Epoch AI specification: } N_{\setminus E}^* \propto C_{\setminus E}^{0.78}, \tag{21}$$

$$\text{Chinchilla specification: } N_{\setminus E}^* \propto C_{\setminus E}^{0.74}, \tag{22}$$

which are close to the Kaplan coefficient of 0.73. **Hence, this shows that the Chinchilla coefficient is roughly consistent with Kaplan's coefficient, given these two adjustments. This constitutes the paper's main result, reconciling these two apparently conflicting results.**

## 4 Experiments: Compute-Parameter Scaling Coefficient

We provide brief experiments verifying that our claims hold for models trained at small scale (millions of parameters).

**Experiment 1.** Firstly, we verify whether scaling coefficients come out close to Chinchilla's and Kaplan's when using $N_T$ and $N_{\setminus E}$ respectively.

We trained five models of sizes, $N_T \in [0.8M, 1.6M, 2.1M, 3.3M, 4.6M]$ on the BookCorpus dataset. We used the GPT-2 tokenizer with vocab size of 50,257, and a context length of 16 (whilst much smaller than typical, our experiments suggest scaling coefficients are not affected by context length). Chinchilla's Method 1 was used to fit scaling coefficients, with the approximation $C = 6ND$.

Models were trained for updates $\in [4000, 4000, 4000, 8000, 8000]$, batchsize was 65,536 tokens per update, for total training tokens $D \in [262M, 262M, 262M, 524M, 524M]$. The best learning rate for each model size was chosen $\in [0.001, 0.005, 0.01, 0.05]$ and no annealing was applied.

**Result 1.** Table 1 shows that when coefficients are fitted to $N_T$, we find $N_T \propto C_T^{0.49}$ and for $N_{\setminus E}$, we find $N_{\setminus E} \propto C_{\setminus E}^{0.74}$. These match closely with the Chinchilla and Kaplan coefficients.

**Experiment 2.** We provide an ablation of optimization schemes demonstrating that using multiple training budgets per model affects coefficients only marginally (counter to Chinchilla's explanation).

- **Scheme 1.** A single learning rate of 0.001 is set for all models. A single model trained per size, and no annealing applied.

- **Scheme 2.** The best learning rate is chosen per model. A single model trained per size, and no annealing applied. (As in our $N_T$ vs. $N_{\setminus E}$ comparison.)

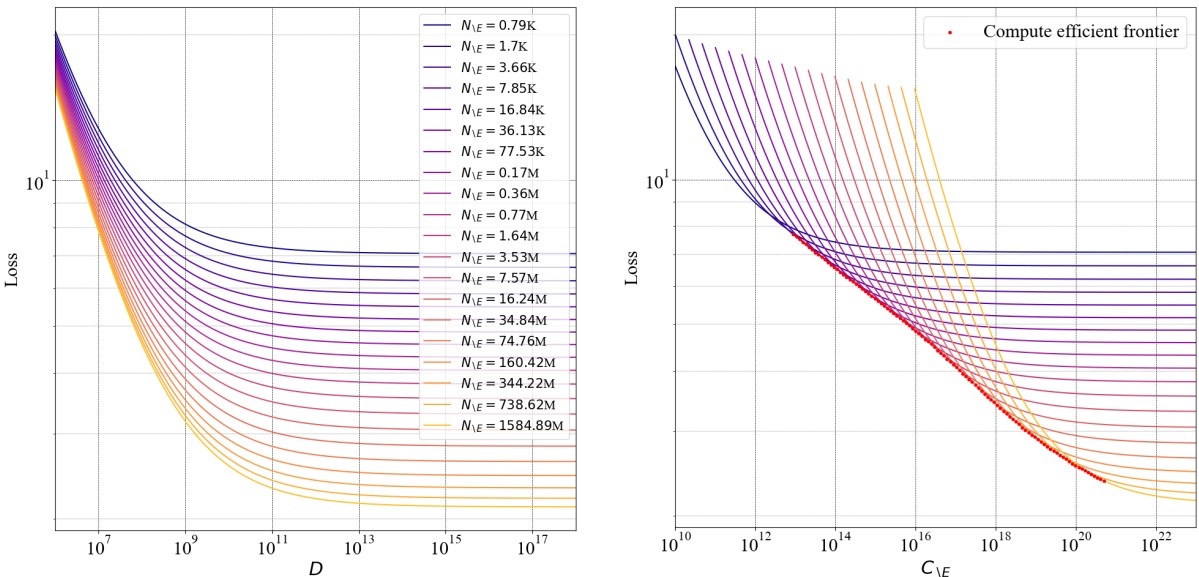

Figure 4: Synthetic training curves from Eq. 18 fitted to the Chinchilla data using the Epoch AI specification. Curves are generated for 20 logarithmically-spaced models matching Kaplan's size range. Left in terms of training tokens, right in terms of non-embedding compute, as used in the Kaplan study. Hence the right plot can be viewed as Chinchilla's loss curves, adjusted to match Kaplan's model sizes and non-embedding measurements.

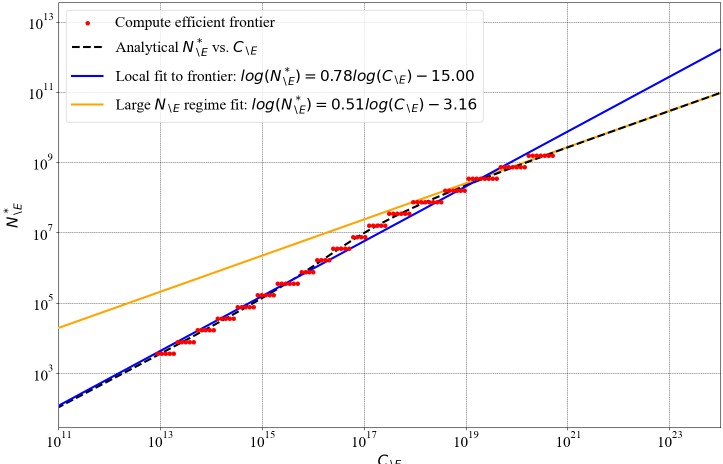

Figure 5: Using the synthetic training curves generated in Figure 4, we empirically fit the frontier of compute efficient models using Chinchilla's Method 1. This gives our main result; synthetic training curves generated from Chinchilla's study, adjusted for model sizes and non-embedding compute, produce a local scaling coefficient $N^*_{\backslash E} \propto C^{0.78}_{\backslash E}$, close to Kaplan's reported coefficient of 0.73. The analytical function from Eq. 19 is also verified.

- **Scheme 3.** The best learning rate is chosen per model. A single model trained per size, and cosine annealing applied at the update budget. (**Kaplan study used this.**)

- **Scheme 4.** The best learning rate is chosen per model. Six models trained per size at different budgets $\in [0.25D, 0.5D, 0.75D, 1.0D, 1.5D, 2.0D]$, and cosine annealing applied. (**Chinchilla study used this.**)

**Result 2.** Table 1 shows that optimization scheme has a smaller impact on scaling coefficients than switching from $N_T$ to $N_{\backslash E}$. Using a single set of models with no annealing (scheme 2) produces the same coefficients as using the more computationally expensive scheme 4. Counter to Chinchilla's comment that moving from Kaplan's scheme 3 to scheme 4 would reduce the scaling coefficient, our experiment suggests the opposite is the case, increasing from 0.46 to 0.49. This might explain our slight overestimation of the scaling coefficients in Eq. 21 & 22.

Table 1: Comparison of different scaling coefficients from our experiments. Note that the change moving from $N_T$ to $N_{\backslash E}$ has a much larger effect than moving between optimization schemes.

| Experiment | $a$ where $N_{\text{optimal}} \propto C^a$ | $b$ where $D_{\text{optimal}} \propto C^b$ |
|---|---|---|
| Chinchilla, $N_T$ | 0.50 | 0.50 |
| Kaplan, $N_{\backslash E}$ | 0.73 | 0.27 |
| | | |
| **Ablating $N_T$ vs $N_{\backslash E}$** | | |
| Ours, $N_T$ & $C_T$ | 0.49 | 0.51 |
| Ours, $N_{\backslash E}$ & $C_{\backslash E}$ | 0.74 | 0.26 |
| | | |
| **Ablating optimization scheme** | | |
| Ours, $N_T$, scheme 1, single lrate, no anneal | 0.58 | 0.42 |
| Ours, $N_T$, scheme 2, best lrate, no anneal | 0.49 | 0.51 |
| Ours, $N_T$, scheme 3, best lrate, single-cosine anneal | 0.46 | 0.54 |
| Ours, $N_T$, scheme 4, best lrate, multi-cosine anneal | 0.49 | 0.51 |

## 5 Analysis: Compute-Loss Scaling Coefficient

As well as analyzing the compute-optimal parameter scaling, Kaplan and Chinchilla also described the scaling relationship between compute and loss, assuming parameters were scaled optimally. This optimal loss was again given in terms of non-embedding compute by Kaplan, and total compute by Chinchilla,

$$L_{\backslash E}^* = \min_{\text{s.t. } C_{\backslash E} = 6N_{\backslash E}D} L(N_{\backslash E}, C_{\backslash E}), \tag{23}$$

$$L_T^* = \min_{\text{s.t. } C_T = 6N_T D} L(N_T, C_T). \tag{24}$$

Concretely, the two studies reported the following forms and coefficients linking optimal loss and compute.

$$\text{Kaplan compute-loss form: } L_{\backslash E}^* = \left( \frac{C_{\backslash E}}{C_0} \right)^{-\gamma} \tag{25}$$

$$\text{Kaplan compute-loss fit: } L_{\backslash E}^* \propto C_{\backslash E}^{-0.057} \tag{26}$$

$$\text{Chinchilla compute-loss form: } L_T^* = \left( \frac{C_T}{C_0} \right)^{-\gamma} + E \tag{27}$$

$$\text{Chinchilla compute-loss fit, Epoch AI spec: } L_T^* - E \propto C_T^{-0.178} \tag{28}$$

$$\text{Chinchilla spec: } L_T^* - E \propto C_T^{-0.155} \tag{29}$$

(See Section A.3 for Chinchilla's compute coefficient.) Similar to the compute-parameter scaling coefficient, on the surface Kaplan's coefficient of 0.057 appears quite far from Chinchilla's of 0.155–0.178. However, we will again show that by beginning from the Chinchilla study, and adjusting for Kaplan's non-embedding compute, smaller scale, and additionally their compute-loss form, these two coefficients can be roughly reconciled.

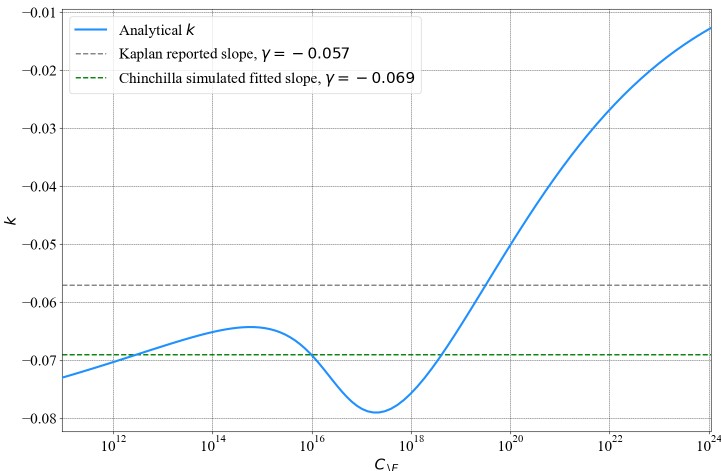

Figure 6: Visualization of Eq. 30, using the Epoch AI specification.

Our analysis follows the same four-step approach as in Section 3. We can directly reuse Steps 1 & 2, while Steps 3 & 4 are now modified to study the relationship between optimal loss and compute, rather than optimal parameters and compute as previously.

**Step 3.** *Analytically derive the relationship between $L^*_{\backslash E}$ & $C_{\backslash E}$.*

We find that the relationship between $L^*_{\backslash E}$ & $C_{\backslash E}$ is not a power law (derived in Section A.2).

$$\frac{d\log(L^*_{\backslash E})}{d\log(C_{\backslash E})} = \frac{g}{L^*_{\backslash E}} \left[ -\alpha N_c \frac{\left( N^*_{\backslash E} + \frac{1}{3}\omega \left( N^*_{\backslash E} \right)^{1/3} \right)}{\left( N^*_{\backslash E} + \omega \left( N^*_{\backslash E} \right)^{1/3} \right)^{\alpha+1}} + \beta D_c \left( \frac{C_{\backslash E}}{6N^*_{\backslash E}} \right)^{-\beta} \left( 1 - \frac{1}{g} \right) \right]. \tag{30}$$

However, again we can consider a local first-order approximation, $L^*_{\backslash E} \propto C^k_{\backslash E}$, where $k = \frac{d\log(L^*_{\backslash E})}{d\log(C_{\backslash E})}$. We visualize this in Figure 6.

**Step 4.** *Simulate synthetic data from the Chinchilla loss model over the model sizes used Kaplan. Fit a local power law, for $L^*_{\backslash E}$ in terms of $C_{\backslash E}$, using Kaplan's compute-loss form.*

As in Section 3, we *could* use Eq. 30 to compute a point estimate for $k$ in the relationship $L^*_{\backslash E} \propto C^k_{\backslash E}$. However, again we opt for the more faithful procedure of simulating data from the loss curves. We then fit Kaplan's compute-loss form $L^*_{\backslash E} = \left( \frac{C_{\backslash E}}{C_0} \right)^{-\gamma}$. For the two specifications, these give the following models,

$$\text{Epoch AI specification:} L^*_{\backslash E} \propto C^{-0.069}_{\backslash E}, \tag{31}$$

$$\text{Chinchilla specification:} L^*_{\backslash E} \propto C^{-0.066}_{\backslash E}, \tag{32}$$

which are roughly inline with Kaplan's reported coefficient of $L^*_{\backslash E} \propto C^{-0.057}_{\backslash E}$.

Figure 7 visualizes the fit of the Chinchilla compute-loss form vs. the Kaplan compute-loss form, when one counts total compute vs. non-embedding compute. We see that Kaplan's form provides a good fit of the data in the non-embedding compute plot at small scale, over the range of model sizes they considered. We speculate that this might be the motivation for Kaplan's selection of this simpler compute-loss form.

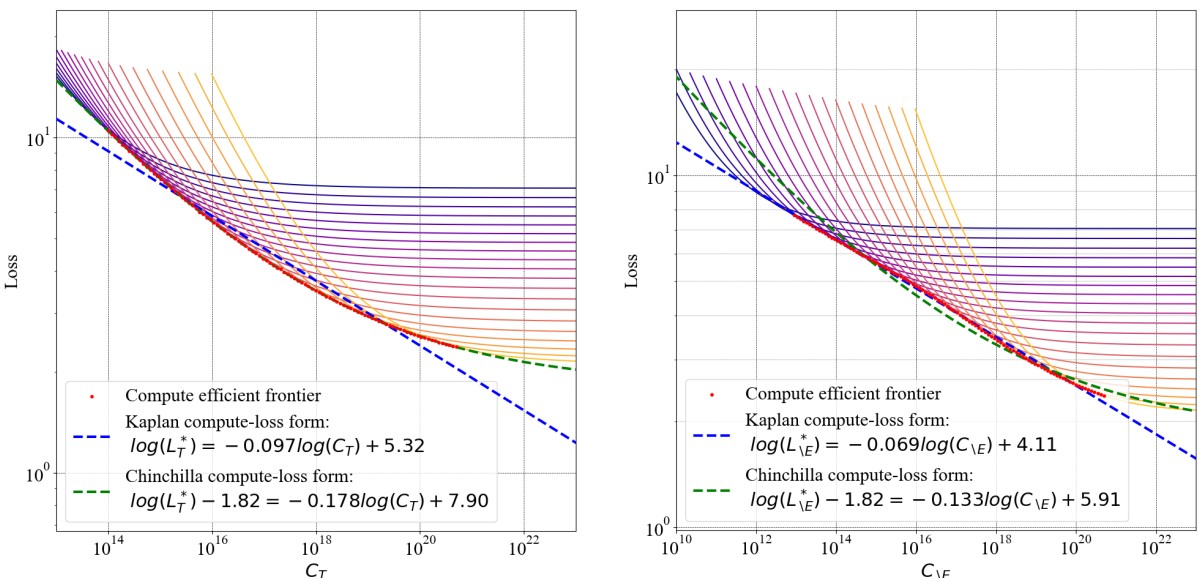

Figure 7: Synthetic data with the Epoch AI specification, as in Figure 4. Here we visualize the quality of the fits using both the Kaplan and Chinchilla compute-loss forms. Notice that Chinchilla's form provides a better fit under total compute, while Kaplan's form is a better match under non-embedding compute.

# 6 Related work

Following early works formalizing how language models improve with parameters, data, and training compute (Rosenfeld et al., 2019; Kaplan et al., 2020; Hoffmann et al., 2022), there has been investigation into whether these scaling laws arise in other domains (Henighan et al., 2020), and to explain their existence from a theoretical standpoint (Hutter, 2021; Maloney et al., 2022; Bahri et al., 2024).

Closer in spirit to our paper are several concurrent works that have investigated the influence of various design decisions on scaling law analyses. Su et al. (2024) revisit the methodology used to find scaling coefficients. Hägele et al. (2024) found that multiple independent cosine schedules could be reproduced more efficiently through a constant learning rate with multiple short decays, or stochastic weight averaging. Our finding is subtly different; a simple fixed learning rate will recover very similar compute-parameter scaling coefficients as multiple cosine schedules. Bi et al. (2024) study the effect of various hyperparameters on scaling laws. They observe that different text datasets produce slightly different optimal coefficients, with 'cleaner' data leading to more parameter-hungry scaling behavior, which they speculate could partially explain the difference between Kaplan and Chinchilla coefficients.

Porian et al. (2024) provide a concurrent work with the same objective as our paper – explaining the differences between the Kaplan and Chinchilla coefficients. Through a set of large-scale experiments reproducing Kaplan's study, they determine that responsibility for the discrepancy can be attributed, in decreasing order of significance, to; 1) Kaplan counting non-embedding rather than total compute. 2) Kaplan using a fixed-length warmup period that was too long for smaller models, making them appear less efficient. 3) Kaplan not fully tuning optimization hyperparameters. We see these findings as complimentary to our own. We have been able to identify the primary 'first-order' reason using *only* information that was publicly available in the two papers, with a fully analytical approach. (Tiny-scale experiments were run post-hoc as verification.) This illustrates the promise of applying mathematical approaches to the empirical science of scaling.

## 7 Discussion

This paper aimed to explain the difference between the Kaplan and Chinchilla scaling coefficients. We found two issues in Kaplan's study that combined to bias their estimated scaling coefficients; their choice to count only non-embedding parameters, and studying smaller sized model sizes. This means there is curvature in the true relationship between $N_{\backslash E}$ & $N_T$ (Figure 5). At larger values of $N_T$, the embedding parameter counts become negligible, $N_T = N_{\backslash E}$, and differences would not arise. Alternatively, had Kaplan studied relationships directly in terms of $N_T$, this issue would also not arise, even at this smaller scale (confirmed by our Experiment 1 finding $N_T \propto C_T^{0.49}$ even for $N_T < 5M$). The form Kaplan used to predict loss from compute further contributed to differences in the reported compute-loss scaling coefficients.

**Inconsistency across scaling studies.** Existing literature on scaling is not consistent in its use of non-embedding vs. total compute. Some studies (Henighan et al., 2020; Gordon et al., 2021; Ghorbani et al., 2021; Fernandes et al., 2023; Hu et al., 2024; Bi et al., 2024; Su et al., 2024) follow Kaplan's approach, using non-embedding parameters or compute, while others (Clark et al., 2022; Que et al., 2024; Wang et al., 2023) adhere to the Chinchilla approach, using total parameters. Our work indicates that this choice can substantially alter scaling exponents, complicating cross-study comparisons. Similarly, the choice of compute-loss equation varies through the literature. Studies such as (Clark et al., 2022; Brown et al., 2020; Smith & Doe, 2024) opt for the Kaplan compute-loss form without offsets. In contrast, (Henighan et al., 2020; Gordon et al., 2021; Fernandes et al., 2023; Hu et al., 2024; Wang et al., 2023) employ the Chinchilla compute-loss form with non-zero offsets. Again, our work suggests that these methodological differences can lead to significant variations in scaling predictions and interpretations.

The lack of a standardized approach in scaling studies risks making comparisons misleading and insights less clear. We see our work as helping to understand certain decisions made in previous studies that should be standardized. Concretely, we advise future studies to report total, rather than non-embedding, parameters, and to include an offset in the compute-loss fitting models. We discuss motivation for these choices below. Furthermore, our initial evidence does not support using multiple cosine decays per model size – we find a single fixed learning rate per model size is sufficient for measuring compute-optimal parameter coefficients.

**Why should embedding parameters contribute to scaling behavior?** Several works evidence that embedding parameters capture meaningful language properties. Word embeddings can be factorized into semantically interpretable factors (even the shallow Word2vec) (Mikolov et al., 2013a;b; Arora et al., 2018). LLMs learn linear embeddings of space and time across scales (Gurnee & Tegmark, 2024). Developing such meaningful embedding structures allows LLMs to perform high-level language operations, such as arithmetic (McLeish et al., 2024). Therefore, if one believes that the embedding layer does more than just 'translate' tokens to a vector of the correct dimension, we see no reason to exclude them in the parameter count.

**Why should a non-zero offset be used in loss-compute predictions?** The Chinchilla compute-loss form with a non-zero offset (Eq. 27), is a more appropriate form from the perspective of statistical learning. This approach accounts for the concept of irreducible risk (Hernandez et al., 2023), which posits a lower bound on achievable loss regardless of model or dataset size. This may arise from various factors: inherent biases or limitations in the learning algorithm, or noise in the original task. As a concrete example in language modeling, the best a model can do for the prediction of the first token in a sequence, is to estimate the marginal distribution of all tokens, which leads to a non-zero loss.

**Limitations.** We acknowledge several limitations of our analysis. We have aimed to capture the primary 'first order' reason for the difference between the Kaplan and Chinchilla scaling coefficients. But there are multiple other differences between the two studies that likely also affect scaling coefficients (Section 6); datasets (Kaplan used OpenWebText2, Chinchilla used MassiveText), transformer details (Kaplan used learnable position embeddings while Chinchilla's were fixed, also differing tokenizers, vocabulary sizes), optimization scheme (Kaplan used scheme 3, Chinchilla scheme 4, also differing warmup schedules), differences in computation counting (Kaplan used $C = 6ND$, Chinchilla's Method 1 & 2 used a full calculation). However, our work suggested these factors impact coefficients in a more minor way.

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

# A  Appendix

## A.1  Derivation of Eq.20

This section derives Eq.20

$$\frac{1}{g} := \frac{d\log(C_{\backslash E})}{d\log(N^*_{\backslash E})} = 1 - \frac{1}{\beta}\frac{N^{*}_{\backslash E}{}^{2/3} + \frac{\omega}{9}}{N^{*}_{\backslash E}{}^{2/3} + \frac{\omega}{3}} + \frac{\alpha+1}{\beta}\frac{N^{*}_{\backslash E}{}^{2/3} + \frac{\omega}{3}}{N^{*}_{\backslash E}{}^{2/3} + \omega}. \tag{33}$$

First note that

$$\frac{d\log(C_{\backslash E})}{d\log(N^*_{\backslash E})} = \frac{d\log(C_{\backslash E})}{dN^*_{\backslash E}}\frac{dN^*_{\backslash E}}{d\log(N^*_{\backslash E})} \tag{34}$$

$$= \frac{d\log(C_{\backslash E})}{dN^*_{\backslash E}}N^*_{\backslash E} \tag{35}$$

Recall the definition of $C_{\backslash E}$ from Eq. 19

$$C_{\backslash E} = 6N^*_{\backslash E}\left(N^*_{\backslash E} + \frac{\omega}{3}N^{*}_{\backslash E}{}^{1/3}\right)^{-\frac{1}{\beta}}\left(N^*_{\backslash E} + \omega N^{*}_{\backslash E}{}^{1/3}\right)^{\frac{1+\alpha}{\beta}}\left(\frac{\beta}{\alpha}\frac{D_c}{N_c}\right)^{\frac{1}{\beta}} \tag{36}$$

$$\log(C_{\backslash E}) = \underbrace{\log(N^*_{\backslash E})}_{\text{term 1}} - \underbrace{\frac{1}{\beta}\log\left(N^*_{\backslash E} + \frac{\omega}{3}N^{*}_{\backslash E}{}^{1/3}\right)}_{\text{term 2}} + \underbrace{\frac{1+\alpha}{\beta}\log\left(N^*_{\backslash E} + \omega N^{*}_{\backslash E}{}^{1/3}\right)}_{\text{term 3}} + \text{const.} \tag{37}$$

where const. does not depend on $N^*_{\backslash E}$. We now can take the derivative of each term.

Derivative of term 1.

$$\frac{d\log(N^*_{\backslash E})}{dN^*_{\backslash E}} = \frac{1}{N^*_{\backslash E}} \tag{38}$$

Derivative of term 2.

$$\frac{1}{\beta}\frac{d\log\left(N^*_{\backslash E} + \frac{\omega}{3}N^{*}_{\backslash E}{}^{1/3}\right)}{dN^*_{\backslash E}} = \frac{1}{\beta}\frac{d\log\left(N^*_{\backslash E} + \frac{\omega}{3}N^{*}_{\backslash E}{}^{1/3}\right)}{dN^*_{\backslash E} + \frac{\omega}{3}N^{*}_{\backslash E}{}^{1/3}}\frac{dN^*_{\backslash E} + \frac{\omega}{3}N^{*}_{\backslash E}{}^{1/3}}{dN^*_{\backslash E}} = \frac{1}{\beta}\frac{1 + \frac{\omega}{9}N^{*}_{\backslash E}{}^{-2/3}}{N^*_{\backslash E} + \frac{\omega}{3}N^{*}_{\backslash E}{}^{1/3}} \tag{39}$$

Derivative of term 3.

$$\frac{\alpha+1}{\beta}\frac{d\log\left(N^*_{\backslash E} + \omega N^{*}_{\backslash E}{}^{1/3}\right)}{dN^*_{\backslash E}} = \frac{\alpha+1}{\beta}\frac{d\log\left(N^*_{\backslash E} + \omega N^{*}_{\backslash E}{}^{1/3}\right)}{dN^*_{\backslash E} + \omega N^{*}_{\backslash E}{}^{1/3}}\frac{dN^*_{\backslash E} + \omega N^{*}_{\backslash E}{}^{1/3}}{dN^*_{\backslash E}} = \frac{\alpha+1}{\beta}\frac{1 + \frac{\omega}{3}N^{*}_{\backslash E}{}^{-2/3}}{N^*_{\backslash E} + \omega N^{*}_{\backslash E}{}^{1/3}} \tag{40}$$

Then assemble all terms and multiply by $N^*_{\backslash E}$ as per Eq. 35.

## A.2 Derivation of compute-loss analytical form in Eq. 30

This section derives $k$, defined as,

$$k = \frac{d\log(L^*_{\backslash E})}{d\log(C_{\backslash E})}. \tag{41}$$

Expanding with the chain rule we find,

$$k = \frac{d\log(L^*_{\backslash E})}{dL^*_{\backslash E}} \frac{dL^*_{\backslash E}}{dN^*_{\backslash E}} \frac{dN^*_{\backslash E}}{d\log(N^*_{\backslash E})} \frac{d\log(N^*_{\backslash E})}{d\log(C_{\backslash E})}, \tag{42}$$

$$= \frac{N^*_{\backslash E}}{L^*_{\backslash E}} \frac{dL^*_{\backslash E}}{dN^*_{\backslash E}} g, \tag{43}$$

where we previously derived $g = \frac{d\log(N^*_{\backslash E})}{d\log(C_{\backslash E})}$ in Eq. 20.

This leaves us with $\frac{dL^*_{\backslash E}}{dN^*_{\backslash E}}$ to find. First note that $L^*_{\backslash E}$ is given by Eq. 18 when the optimal model size is used, i.e., $N_{\backslash E} \leftarrow N^*_{\backslash E}$,

$$L^*_{\backslash E} = \frac{N_c}{(N^*_{\backslash E} + \omega N^{*\,1/3}_{\backslash E})^\alpha} + \frac{D_c}{(C_{\backslash E}/6N^*_{\backslash E})^\beta} + E. \tag{44}$$

Before taking this derivative, we recall that $C_{\backslash E}$ is actually a function of $N^*_{\backslash E}$ (via Eq. 19). Hence, we tackle the derivative in two parts. We find the first term derivative is equal to,

$$\frac{d\frac{N_c}{(N^*_{\backslash E}+\omega N^{*\,1/3}_{\backslash E})^\alpha}}{dN^*_{\backslash E}} = -\alpha N_c \frac{\left(1 + \frac{1}{3}\omega\left(N^*_{\backslash E}\right)^{-2/3}\right)}{\left(N^*_{\backslash E} + \omega\left(N^*_{\backslash E}\right)^{1/3}\right)^{\alpha+1}}. \tag{45}$$

The derivative of the second term, via the product rule, and spotting that $\frac{dC_{\backslash E}}{dN^*_{\backslash E}} = \frac{C_{\backslash E}}{gN^*_{\backslash E}}$, equals,

$$\frac{d\frac{D_c}{(C_{\backslash E}/6N^*_{\backslash E})^\beta}}{dN^*_{\backslash E}} = \beta D_c \frac{1}{N^*_{\backslash E}} \left(\frac{C_{\backslash E}}{6N^*_{\backslash E}}\right)^{-\beta} - \beta D_c \frac{1}{gN^*_{\backslash E}} \left(\frac{C_{\backslash E}}{6N^*_{\backslash E}}\right)^{-\beta} \tag{46}$$

$$= \beta D_c \frac{1}{N^*_{\backslash E}} \left(\frac{C_{\backslash E}}{6N^*_{\backslash E}}\right)^{-\beta} \left(1 - \frac{1}{g}\right). \tag{47}$$

Hence, combining these two terms we find,

$$\frac{dL^*_{\backslash E}}{dN^*_{\backslash E}} = -\alpha N_c \frac{\left(1 + \frac{1}{3}\omega\left(N^*_{\backslash E}\right)^{-2/3}\right)}{\left(N^*_{\backslash E} + \omega\left(N^*_{\backslash E}\right)^{1/3}\right)^{\alpha+1}} + \beta D_c \frac{1}{N^*_{\backslash E}} \left(\frac{C_{\backslash E}}{6N^*_{\backslash E}}\right)^{-\beta} \left(1 - \frac{1}{g}\right). \tag{48}$$

Combining this result into to Eq. 43 we get,

$$k = g \frac{N^*_{\backslash E}}{L^*_{\backslash E}} \frac{dL^*_{\backslash E}}{dN^*_{\backslash E}} \tag{49}$$

$$= \frac{g}{L^*_{\backslash E}} \left[ -\alpha N_c \frac{\left(N^*_{\backslash E} + \frac{1}{3}\omega\left(N^*_{\backslash E}\right)^{1/3}\right)}{\left(N^*_{\backslash E} + \omega\left(N^*_{\backslash E}\right)^{1/3}\right)^{\alpha+1}} + \beta D_c \left(\frac{C_{\backslash E}}{6N^*_{\backslash E}}\right)^{-\beta} \left(1 - \frac{1}{g}\right) \right]. \tag{50}$$

### A.3 Compute-loss coefficient derivation

We know from Eq. 17 $N_T^* \propto C^{\frac{\beta}{\alpha+\beta}}$, and similarly $D_T^* \propto C^{\frac{\alpha}{\alpha+\beta}}$. Substituting these into the loss form of Eq. 8, and for some new constants $\bar{N}_c, \bar{D}_c$ we find,

$$L_T^* = \frac{N_c}{N_T^{*\,\alpha}} + \frac{D_c}{D_T^{*\,\beta}} + E, \tag{51}$$

$$= \frac{\bar{N}_c}{C^{\frac{\alpha\beta}{\alpha+\beta}}} + \frac{\bar{D}_c}{C^{\frac{\alpha\beta}{\alpha+\beta}}} + E, \tag{52}$$

$$L_T^* - E \propto C^{\frac{-\alpha\beta}{(\alpha+\beta)}}. \tag{53}$$

