# OpenReview forum: "Reconciling Kaplan and Chinchilla Scaling Laws"
_TMLR — Accepted by TMLR_

### Review · Reviewer_Cahx · 2024-10-07

**Summary Of Contributions:**

This paper provides analytical and experimental evidence to reaffirm the Chinchilla’s scaling law. To do so, it shows that the other popular scaling law (Kaplan’s scaling law), which is in contradiction to Chincilla’s scaling law, can indeed be obtained from Chinchilla's scaling law by accounting for the number of embedding parameters. The paper starts with Chinchilla’s scaling law, and using analysis it shows that by simply removing the number of embedding parameters, one obtains Kaplan’s law for small-sized models, which leads to the issue with Kaplan’s scaling law. The paper also provides experimental evidence to back their claims showing i) the two different scaling laws for small-sized models when considering the embedding parameters vs. not considering the embedding parameters, (ii) it also shows that different optimization hyperparameters have little influence on the scaling laws.

**Audience:**

Yes

**Claims And Evidence:**

Yes

**Requested Changes:**

- Please see the weaknesses.

**Strengths And Weaknesses:**

Strengths:
- The paper is well written. The analysis is written in clear steps, making it easy to follow.
- The paper provides evidence for the correctness of the Chinchilla’s scaling law by showing how the two seemingly different scaling laws are related using analysis.
- The paper provides very good theoretical analysis showing the connection between the two scaling laws.
- The paper also provides experimental evidence showing the correctness of Chinchilla’s scaling law, the connection between the two scaling laws, and how optimization hyperparameters do not affect the scaling laws much.
- The paper addresses a very important topic and will help designing large scale models in the future.

Weaknesses:
- This analysis in the paper is independent of the nature of the architecture used, as long as the parameters are given. Is there any insight on whether architecture has any role to play in the scaling laws and whether any further theoretical/experimental analysis can help understand whether the scaling laws apply to architectures beyond the transformers.
- Fig 5 is a bit difficult to understand: I suggested improving the coloring scheme (the similar colors used for the lines are difficult to distinguish), and making the caption and the “Main result” in Section 3 more detailed since it is the core message of the paper and it would help the readability of the paper.
- Since this is an analytical study of scaling laws, it would be better to write down the summary of approximations and assumptions made in the analysis. This would help readers understand how to modify the analysis in the future if the assumptions change, hence, it would avoid the mistake made in Kaplan’s scaling law (their assumption of ignoring the embedding parameters and working with small scale). This would also help because the assumptions and approximations made during the analysis steps can sometimes be hard to follow.
- Minor: Page 4 in step 2, “we need an analytical from of loss” -> “we need an analytical form of loss”

---

> ### Author Response · Authors · 2024-10-28
> **Response**
>
> Thank for your thoughtful review. We are pleased that we were able to successfully communicate the value of our work and no major technical issues were found. We respond to the listed weaknesses below -- see the newly uploaded version to see the changes in the pdf. Please let us know if these satisfy your points.
>
> - __Architecture insights.__ We agree that an understanding of the effect of architecture on scaling laws is an important topic. We don't feel our current work is able to offer particularly new insights into this, since we build upon two studies that themselves were restricted to transformer architectures.
> We would like to explore this in future work though -- we question whether simply counting parameters is the best proxy for model capacity. Potentially connecting to Figure 5 of Kaplan et al. would be an interesting starting point?
>
> - __Readability.__ Thanks for suggesting these improvements. We have changed the color scheme of Figure 5, added more text to the __main result__ section, and caption of Figures 4 and 5.
>
> - __Assumptions.__ We appreciate this suggestion and apologize for not including it in the original version. We have added Assumptions in Section 2.3, which we believe captures the salient assumptions made in the analysis. Please let us know if you feel anything is missing.
>
> - __Minor.__ Fixed.

---

> > ### Comment · Reviewer_Cahx · 2024-11-14
> > **Thanks for the response**
> >
> > I thank the authors for improving the clarity of the work. Thanks for answering the question related to architecture insight, yes, I agree since the prior works are focused on transformers, it is difficult to comment much on other architectures without more study.
> >
> > The authors' response clarifies all my questions.

---

### Review · Reviewer_nDnt · 2024-10-14

**Summary Of Contributions:**

The main goal of the paper is to explain the discrepancy between Kaplan and Chinchilla scaling laws via a theoretical model. The main result shows that sampling from the Chinchilla scaling law, and computing only non-embedding parameters, it is possible to match closely the scaling law from Kaplan et al.

**Audience:**

Yes

**Claims And Evidence:**

Yes

**Requested Changes:**

Minor comments:
* missing space in equation 5 and 6
* page 4, above (12): "from" should be "form"
* it might be helpful for the reader to include a few extra steps for the calculations on page 5 (possibly in the appendix)

**Strengths And Weaknesses:**

In short, I could not find any notable weaknesses in this submission.
The author's hypothesis is that most of the discrepancy can be explained by accounting for embedding parameters in Kaplan's model. Given the widespread use of scaling laws, it is an important study, as the two discussed papers end up with different scaling law exponents.

All steps and derivations are well motivated and explained, and the main result looks very convincing.

The only question I have for the authors is whether they are planning to release the code for the paper (which I would highly encourage).

---

> ### Author Response · Authors · 2024-10-28
> **Response**
>
> Thank you for your comments. We are pleased that the impact of the work has been recognized, and the technical details have held up to your evaluation. We respond below to the points you highlighted. Please let us know if you spot any other areas that could be improved.
>
> - __Code.__ We indeed plan to release code to support the analysis sections of our work.
>
> - __Typos.__ Typos have been addressed in the newly uploaded version.
>
> - __Derivation details.__ We have added a new section in Appendix A.1 with the derivation of Eq. 20 (previously Eq. 17) -- we agree that its derivation was not trivial and apologize for its original exclusion! We have also added a more explicit description of how we reach Eq. 17 and 19 in the main text. Please see the new version of the paper for these changes.

---

### Review · Reviewer_GNQy · 2024-10-14

**Summary Of Contributions:**

Kaplan and Chinchilla presented different coefficients for how the number of parameters (N) should be scaled to achieve the lowest possible loss given a fixed compute budget (C). Chinchilla suggested scaling the amount of data and number of parameters in proportion, while Kaplan suggested to focus more on increasing the number of parameters in contrast to data.

This paper attempts to understand this discrepancy between the two scaling laws. In particular, the paper highlights that two fundamental reasons for this difference: (i) Kaplan focused on non-embedding parameters, while Chinchilla focused on total parameters, and (ii) Kaplan focused on smaller models (1k to 1.5B parameters) while Chinchilla considered larger models (44M to 16B).

While other differences exist such as differences in embedding types and dataset used for training, the paper shows that they only marginally impact the conclusions.

**Audience:**

Yes

**Claims And Evidence:**

Yes

**Requested Changes:**

The reviewer would suggest a proper background section that sets the stage for the whole paper, without requiring the reader to be very familiar with the exact parameterizations of scaling laws. This can significantly improve the accessibility of the paper for a wider audience.

**Strengths And Weaknesses:**

### Strengths:
- The paper attempts to reconcile two fundamental scaling laws and highlights the reasons for the discrepancy between the obtained coefficients.
- The paper reanalyzes the data based on these assumptions, highlighting that Chinchilla would also obtain similar coefficients when focusing only on non-embedding parameters, and considering smaller model sizes.
- The paper presents toy experiments and highlight the conclusions hold in practice.

### Weaknesses
- While the paper is technically solid and interesting, the reviewer felt that the paper missed important background that is important to understand the paper. Therefore, the reviewer would suggest including a proper background section that describes all the relevant details from Kaplan and Chinchilla papers in order to make the paper more understandable and standalone such as describing the L(N, D) or L(N, C) parameterizations directly within the background.

---

> ### Author Response · Authors · 2024-10-28
> **Response**
>
> Thank you for taking the time to review our work. It's encouraging that we have been able to communicate the impact of our work clearly, and have avoided any technical concerns.
>
> We agree that the previous version of the paper would be a little challenging for readers unfamiliar with the Kaplan and Chinchilla studies. We have incorporated your feedback, creating a significantly longer __preliminaries__ section in the new version of the paper. This provides a more gradual introduction to power laws, and definitions of parameter counting, compute optimality and Chinchilla's functional form.
>
> We believe this has improved the readability of the paper significantly and hope it has addressed your concern. Let us know if you'd like any further background included.

---

> > ### Comment · Reviewer_GNQy · 2024-11-01
> > **Thanks for the update**
> >
> > Thanks for taking the time to update the paper. I certainly think that this change has improved the overall readability of the paper and the flow of ideas. I am happy with the change.

---

### Decision · Action_Editor_jTPV · 2024-11-17

**Recommendation:** Accept as is

**Comment:**

The paper investigates the discrepancy between previous studies on scaling "laws" by Kaplan and the Chinchilla paper, who reach different conclusions w.r.t. the optimal parameter count per training set size. The paper demonstrates convincingly that the discrepancy can be explained by the lack of counting embedding parameters in Kaplan, and the experiments mainly done on smaller architectures by Kaplan, which introduces a bias away from Chinchilla's results. This clears up a long-standing discrepancy in a clear and comprehensive fashion. While this contribution is worthy of a publication in itself, the execution of the paper (quality of the writing and experimental evaluation) is very good. In addition to meticulously performing Chinchilla-style analysis under Kaplan's settings (with small models and on BookCorpus), the paper provides a comprehensive theoretical analysis and has some important insights and best-practices for (future) scaling analysis for the field (e.g., focus on total parameters and total compute vs. compute-loss analysis).

Overall, I think the paper is a clear accept. I also recommend the paper to receive a Reproducibility Certification - even though, it does not precisely repeat a previously reported study, it analyzes and clearly explains how previously reported results from two studies arise, and how a reproduction of one of the previous studies (Chinchilla on Kaplan's setting) resolves the discrepancy.

**Audience:**

All reviewers and I agree that the paper is interesting to a substantial part of the TMLR audience.

**Claims And Evidence:**

All reviewers agree that the paper makes clear and falsifiable claims that are well supported by the evidence presented in the paper, and I agree with that.